# ScaleMoR: Multi-Scale Mixture of Recursive Linear Experts for Time Series Forecasting

## Abstract

Multivariate time-series forecasting across multiple horizons faces two major challenges: temporal misalignment when aggregating multi-scale representations, and inefficient uniform computation regardless of sequence complexity. Existing methods either lose temporal dependencies during multi-scale fusion or allocate computation uniformly while ignoring varying input characteristics, which reduces long-range forecasting performance. We propose ScaleMoR, a novel architecture that redefines mixture-of-experts for temporal modeling. ScaleMoR applies recursive, scale-specific linear transformations as "experts", enabling parameter-efficient conditional computation. The method introduces three key innovations: (1) temporal-aligned multi-scale tokenization, which preserves chronological consistency across fine (2-step), medium (6-step), coarse (12-step), and macro (24-step) windows using learned Gaussian-weighted interpolation, (2) multi-dimensional complexity routing, which dynamically allocates computation based on trend, seasonal, noise, and volatility characteristics instead of a single complexity measure; and (3) hierarchical recursive modules, where deeper layers employ SwiGLU gating and dilated convolutions to achieve progressively richer representations through linear operations alone. We adopt a progressive three-phase training strategy that first learns tokenization, then introduces routing with entropy regularization, and finally optimizes the full architecture. Across ten benchmark datasets and multiple forecast horizons, ScaleMoR consistently outperforms state-of-the-art models, with particularly strong gains on long-range prediction tasks. It delivers 75–85% fewer parameters and 96–99% fewer FLOPs compared to recent attention-based and clustering-based approaches, while maintaining or surpassing their accuracy. These results highlight ScaleMoR as a highly accurate and efficient solution for multivariate time-series forecasting, well suited to real-world domains such as finance, energy, and industrial monitoring.

**Keywords:** Time Series Forecasting, Mixture-of-Experts, Recursive Neural Networks, Multi-Scale Representation

## 1 Introduction

Multivariate time-series (MTS) forecasting is a long-standing challenge in machine learning, with applications in finance Sezer et al. (2020), energy Zhang et al. (2018), weather Schultz et al. (2021), and healthcare Faust et al. (2018). The difficulty lies in capturing temporal dependencies across multiple scales while allocating computation efficiently based on sequence complexity. Transformer-based models have shown promise Nie et al. (2023); Liu et al. (2024), but their quadratic cost and susceptibility to temporal misalignment limit performance on long-range tasks.

A common strategy for multi-scale modeling is to process signals at different resolutions in parallel and then fuse them via concatenation or attention Wu et al. (2021); Zhou et al. (2022a). However, combining fine-grained patterns such as hourly dynamics with coarse-grained trends such as monthly cycles without explicit temporal alignment can break chronological consistency, for example through misaligned aggregation windows or phase shifts, which degrades the resulting representations Wang et al. (2024). Recent methods attempt to address this challenge by clustering temporal and channel patterns Li et al. (2024b) or by introducing explicit alignment mechanisms Jin et al. (2024b), but these approaches often introduce substantial architectural complexity or computational

overhead. Wavelet-based mixture-of-experts (MoE) has also been explored Zhang et al. (2024), yet this design raises parameter counts and inherits the inefficiency of independently trained experts.

Other directions include state-space models Ansari et al. (2024), transformer variants with specialized attention patterns Cao et al. (2024), and linear methods that often rival more complex designs Zeng et al. (2023). Yet most approaches still allocate computation uniformly to all tokens regardless of complexity Kitaev et al. (2020), wasting resources when simple regularities are processed with the same budget as highly variable dynamics.

The MoE paradigm offers a way to introduce conditional computation Shazeer et al. (2017); Fedus et al. (2022). Traditional MoE models employ multiple independent networks as experts and use a gating function to route inputs Jacobs et al. (1991). While this has been successful in natural language processing (NLP) Lepikhin et al. (2021) and computer vision (CV) Riquelme et al. (2021), its use in time series forecasting is limited Chen et al. (2024). Recent wavelet-based MoE methods Zhang et al. (2024) attempt to capture multi-scale dynamics but still suffer from parameter inefficiency and lack explicit alignment across temporal scales. We argue that independent parallel experts are fundamentally suboptimal for time series, where sequential dependencies and efficient computation are essential.

In this work, we introduce *ScaleMoR* (Scale-aware Mixture of Recursive experts), which fundamentally redefines the MoE paradigm for MTS forecasting. Instead of employing independent parallel experts, we treat different recursive application depths of shared linear transformations as "experts." This formulation offers three key advantages over traditional MoE approaches Zhang et al. (2024); Jacobs et al. (1991): (1) *parameter efficiency* through weight sharing across recursive depths, (2) *computational efficiency* via pure linear operations without attention mechanisms Zeng et al. (2023), and (3) *interpretability* where expert selection corresponds to processing depth allocation.

We introduce ScaleMoR (Scale-aware Mixture of Recursive experts), a new architecture that redefines MoE for time series forecasting. Instead of using independent parallel experts, we interpret different recursive application depths of shared linear transformations as "experts." This design brings three advantages: (1) parameter efficiency through weight sharing, (2) computational efficiency through linear operations without attention Zeng et al. (2023), and (3) interpretability, since expert choice reflects the depth of recursive processing.

ScaleMoR also addresses temporal misalignment through a Gaussian-weighted interpolation mechanism that produces aligned tokens at four scales. This avoids both the phase inconsistency of naïve fusion and the complexity of clustering Li et al. (2024b) or wavelet decomposition Zhang et al. (2024). We further introduce multi-dimensional complexity assessment that evaluates each token's trend, seasonality, noise, and volatility, enabling finer-grained allocation of computation than scaler measures Wu et al. (2021). Finally, hierarchical recursive modules progressively expand representational capacity through SwiGLU gating Shazeer (2020) and dilated convolutions Yu & Koltun (2016), achieving strong modeling power without the cost of attention. To ensure stable training, ScaleMoR employs a progressive three-phase training strategy inspired by curriculum learning Bengio et al. (2009): (1) tokenization learning with frozen routing, (2) routing with entropy regularization Pereyra et al. (2017), and (3) full architecture optimization. This staged process prevents routing from interfering with representation learning and supports convergence.

Our contributions are threefold. First, we introduce a conceptual innovation by redefining the MoE paradigm for temporal modeling through recursive depth sharing, improving parameter efficiency and interpretatbility. Second, we advance multi-scale forecasting with temporal-aligned tokenization and multi-dimensional complexity routing, resolving misalignment and enabling adaptive computational allocation. Third, we empirically validate ScaleMoR across ten benchmark datasets and multiple forecast horizons, demonstrating state-of-the-art performance while achieving substantial efficiency gains over recent transformer-based Jin et al. (2024a); Liu et al. (2024), clustering-based Li et al. (2024b), and MoE architectures Zhang et al. (2024).

## 2 RELATED WORK

Transformer architectures have become prominent in MTS forecasting. Informer Zhou et al. (2021) introduced ProbSparse attention to mitigate the quadratic cost of long-sequence modeling, while Autoformer Wu et al. (2021) employed decomposition with auto-correlation to capture periodic

patterns. FEDformer Zhou et al. (2022a) extended this line by modeling seasonality with frequency-enhanced attention in the Fourier domain. More recent efforts have focused on patching strategies for efficiency. For example, PatchTST Nie et al. (2023) treats patches of time series as tokens, achieving strong performance through channel independence and effective patching, while iTransformer Liu et al. (2024) inverts the embedding process by applying attention across the variate rather than the temporal dimension, improving multivariate forecasting. Despite these advances, transformer-based methods still suffer from temporal misalignment when integrating multi-scale patterns and allocate computation uniformly across tokens, regardless of complexity.

In parallel, simple non-transformer architectures have proven competitive. Linear models such as DLinear Zeng et al. (2023) and RLinear Li et al. (2024a) show that decomposition-based linear layers with proper normalization can rival or surpass transformers. Beyond linear methods, CNN-based approaches have gained traction. TimesNet Wu et al. (2022) projects 1D time series into 2D tensors to exploit CV techniques for multi-periodicity, while DUET Li et al. (2024b) integrates autoencoders with dual-stage clustering for temporal and channel dependencies. While effective, these methods typically apply uniform computation across all temporal positions, limiting efficiency.

Another line of research explores MoE architectures to scale models through conditional computation. Switch Transformer Fedus et al. (2022) introduced token-choice routing, assigning each token to one expert, while GLaM Du et al. (2022) demonstrated the efficiency of sparse expert activation. Expert Choice Routing Zhou et al. (2022b) reversed the scheme, improving load balancing by letting experts select tokens. MoE has also evolved toward recursive designs. The Mixture-of-Recursions framework Bae et al. (2025) applied shared layers recursively with dynamic depth, offering parameter efficiency without sacrificing capacity. However, most of these methods have been validated in NLP and CV, with limited exploration in MTS forecasting, where temporal alignment and scale-awareness remain critical.

Related to this, multi-scale temporal modeling has long been a challenge. Classical approaches rely on parallel pathways for different resolutions Lai et al. (2018); Salinas et al. (2020), but naïve concatenation often introduces temporal misalignment. SCINet Liu et al. (2022) addressed this with recursive splitting and cross-scale interaction, while Triformer Cirstea et al. (2022) employed triangular attention to model variable-wise and temporal dependencies simultaneously. More adaptive methods have emerged as well. For instance, TimeMixer Wang et al. (2024) alleviates misalignment via decomposable mixing blocks, though it still applies uniform computation. Whereas ModernTCN Luo & Wang (2024) shows that well-designed temporal convolutions can match transformer accuracy with greater parameter efficiency.

Finally, efficiency-oriented work has investigated conditional computation, gating, and activation mechanisms. AdaViT Xing et al. (2022) introduced adaptive token processing in vision transformers, allowing different regions to receive variable computation, while DynaBERT Hou et al. (2020) applied adaptive width selection for efficiency in NLP. In MTS forecasting, however, conditional computation remains underexplored. Most approaches adapt at the input-level Kitaev et al. (2020) or use uniform expert allocation Chen et al. (2024), overlooking the varying complexity of temporal patterns. Gating and activation function advances provide further inspiration. SwiGLU Shazeer (2020), combining Swish activation with gating, has become standard in large-scale models due to improved gradient flow, while earlier GLU variants Dauphin et al. (2017) also exhibited benefits for sequential dependencies. Similarly, dilated convolutions, first introduced in WaveNet Van Den Oord et al. (2016), efficiently expand receptive fields and have been successfully adapted to MTS forecasting Bai et al. (2018), aligning naturally with the hierarchical structure of temporal data.

## 3 METHODOLOGY

ScaleMoR consists of four main components that work synergistically to achieve efficient and accurate MTS forecasting: (1) a Temporal-Aligned Tokenizer that extracts multi-scale temporal representations while preserving chronological consistency, (2) an Enhanced Multi-Dimensional Router that adaptively allocates computational resources based on input complexity characteristics, (3) Cross-Scale Positional Encoding that captures inter-scale temporal relationships, and (4) an Adaptive Prediction Head that adjusts its architecture based on forecast horizon requirements.

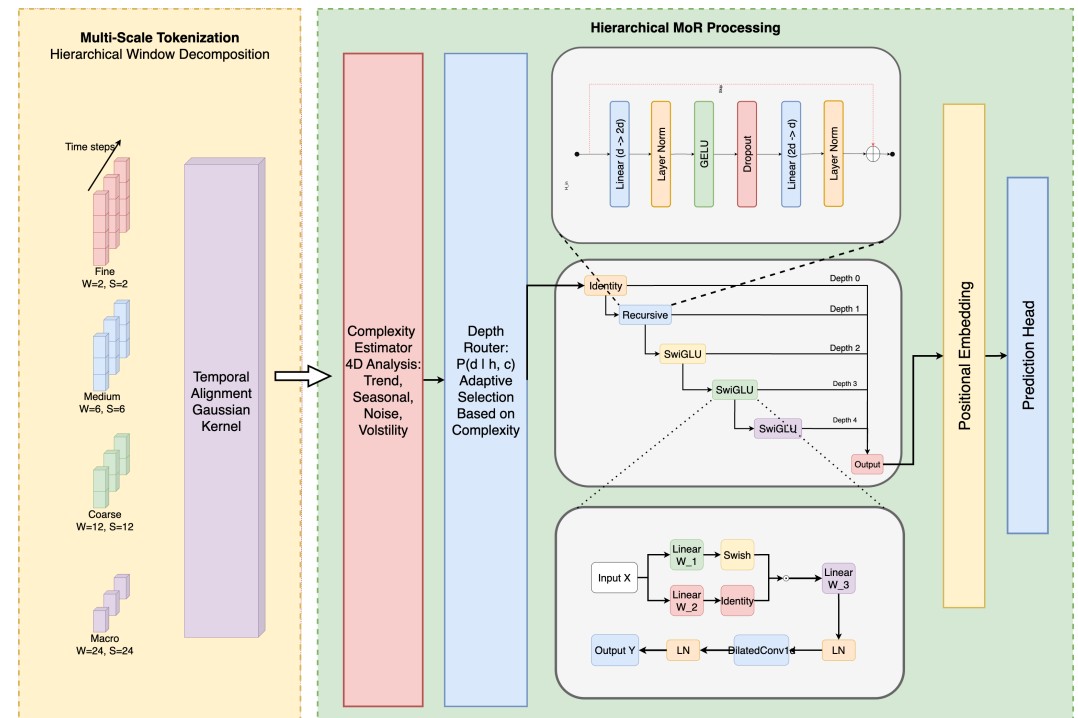

Figure 1: Overall ScaleMoR Architecture Diagram

### 3.1 TEMPORAL-ALIGNED MULTI-SCALE TOKENIZATION

Traditional multi-scale approaches often suffer from temporal misalignment when aggregating representations across scales. Our temporal-aligned tokenization maintains chronological consistency during fusion. We define four scales: Fine (2-step window/stride, local variations), Medium (6-step, short-term patterns), Coarse (12-step, medium-term trends), and Macro (24-step, long-term patterns).

For scale $s$ with window $w_s$ and stride $st_s$, tokens are extracted as $\mathbf{T}_i^{(s)} = \text{Pool}^{(s)}(\mathbf{X}_{i \cdot st_s : i \cdot st_s + w_s})$, where $\text{Pool}^{(s)}$ is max (fine), mean (medium), or attention-weighted (coarse/macro).

Temporal consistency is ensured by Gaussian interpolation: $\mathbf{W}_{i,t}^{(s)} = \exp\left(-\frac{(t - c_i^{(s)})^2}{2\sigma^2}\right) / Z_i^{(s)}$, with $c_i^{(s)}$ the token center, $\sigma$ the radius, and $Z_i^{(s)}$ the normalizer.

Each scale is projected to a shared space via $\mathbf{H}^{(s)} = \text{LayerNorm}\left(\text{ReLU}(\mathbf{W}_1^{(s)}\mathbf{T}^{(s)} + \mathbf{b}_1^{(s)})\right) + \mathbf{P}^{(s)}$, where $\mathbf{P}^{(s)}$ are learnable positional embeddings.

### 3.2 MULTI-DIMENSIONAL COMPLEXITY ROUTING

Traditional MoE models often rely on single-scalar complexity measures. Our multi-dimensional routing mechanism evaluates four distinct complexity dimensions to make more informed computational allocation decisions.

We design a complexity estimator that extracts four fundamental characteristics from each token:

$$\mathbf{C} = \sigma(\mathbf{W}_C \text{ReLU}(\mathbf{W}_{C1}\mathbf{h} + \mathbf{b}_{C1}) + \mathbf{b}_C),  \tag{1}$$

where $\mathbf{C} = [c_{\text{trend}}, c_{\text{seasonal}}, c_{\text{noise}}, c_{\text{volatility}}]^T$ is the four-dimensional complexity vector, and $\sigma$ ensures scores remain in $[0, 1]$.

Different temporal scales naturally favor different processing depths. We introduce learnable scale-specific bias parameters:

$$\mathbf{B}^{(s)}_{\text{trend}} = \text{Parameter}([\beta_0^{(s)}, \ldots, \beta_D^{(s)}]), \quad \mathbf{B}^{(s)}_{\text{seasonal}} = \text{Parameter}([\gamma_0^{(s)}, \ldots, \gamma_D^{(s)}]),$$

where $D$ is the maximum recursive depth. Fine-scale tokens favor shallow processing ($\beta_0^{(\text{fine})} > \beta_D^{(\text{fine})}$), while macro-scale tokens favor deeper recursive processing ($\beta_D^{(\text{macro})} > \beta_0^{(\text{macro})}$).

Routing combines token representations with complexity features to determine optimal processing depths:

$$\mathbf{R} = \text{Softmax}\left(\mathbf{W}_R[\mathbf{h}; \mathbf{C}] + c_{\text{trend}}\mathbf{B}^{(s)}_{\text{trend}} + c_{\text{seasonal}}\mathbf{B}^{(s)}_{\text{seasonal}}\right), \tag{2}$$

where $[\mathbf{h}; \mathbf{C}]$ denotes the concatenation of token features and complexity scores, and $\mathbf{R} \in \mathbb{R}^{D+1}$ represents the probability distribution over recursive depths $\{0, 1, \ldots, D\}$.

## 3.3 HIERARCHICAL RECURSIVE PROCESSING MODULES

Hierarchical recursive modules replace attention mechanisms, providing progressive representational capacity via linear operations with gating and dilated convolutions.

Depth-stratified recursive modules feature shallow and deep paths, each tailored to its function.

Shallow Modules (Depths 1) focus on rapid, lightweight transformations:

$$\mathbf{h}^{(d)} = \text{LayerNorm}\left(\text{GELU}\left(\mathbf{W}_2 \text{LayerNorm}\left(\text{GELU}(\mathbf{W}_1\mathbf{h}^{(d-1)})\right)\right)\right).$$

Deep Modules (Depths 2+) employ SwiGLU gating and dilated convolutions for enhanced representational power:

$$\mathbf{g}^{(d)} = \text{SiLU}(\mathbf{W}_g\mathbf{h}^{(d-1)}) \odot (\mathbf{W}_v\mathbf{h}^{(d-1)}), \quad \mathbf{h}^{(d)}_{\text{swiglu}} = \mathbf{W}_o\mathbf{g}^{(d)},$$

$$\mathbf{h}^{(d)}_{\text{conv}} = \text{DilatedConv}(\mathbf{h}^{(d)}_{\text{swiglu}}, \text{dilation} = 2^{d-2}), \quad \mathbf{h}^{(d)} = \text{LayerNorm}(\mathbf{h}^{(d)}_{\text{conv}}),$$

where $\odot$ denotes element-wise multiplication, and the dilation factor grows exponentially with depth.

Each token's final representation is

$$\mathbf{h}_{\text{final}} = \sum_{d=0}^{D} r_d \cdot \mathbf{f}^{(d)}(\mathbf{h}_{\text{input}}), \quad \mathbf{f}^{(d)}(\mathbf{h}) = \begin{cases} \mathbf{h}, & d = 0, \\ \mathbf{h} + \alpha \cdot \mathcal{M}_d(\mathbf{f}^{(d-1)}(\mathbf{h})), & d > 0, \end{cases}$$

with $\alpha = 0.5$ serving as a residual scaling factor.

## 3.4 CROSS-SCALE POSITIONAL ENCODING

To model relationships across temporal scales, we introduce cross-scale relative positional encodings that represent interactions between tokens from different scales.

For each pair of scales $(s_i, s_j)$ with $i < j$, we define relative positional encodings:

$$\mathbf{E}^{(s_i, s_j)} = \text{Parameter}\left(\mathbb{R}^{1 \times 1 \times d_{\text{model}}}\right).$$

The cross-scale enhancement for tokens at scale $s$ is computed as:

$$\mathbf{H}^{(s)}_{\text{enhanced}} = \mathbf{H}^{(s)} + \sum_{s' \neq s} \text{sign}(s, s') \cdot \mathbf{E}^{(\min(s,s'), \max(s,s'))},$$

where $\text{sign}(s, s')$ ensures consistent directionality in the relative position relationships.

## 3.5 Adaptive Prediction Head

Our adaptive prediction head dynamically adjusts its structure according to the prediction length, accounting for the distinct characteristics of short-, medium-, and long-term forecasts.

Prediction tasks are grouped into three categories: Short-term ($L \leq 96$) uses a 3-layer MLP emphasizing detail preservation; Medium-term ($96 < L \leq 336$) employs a 4-layer architecture balancing detail and trend modeling; Long-term ($L > 336$) incorporates TrendConv layers for trend extraction:

$$\mathbf{y} = \begin{cases} \mathbf{W}_{s3} \operatorname{ReLU}(\mathbf{W}_{s2} \operatorname{GELU}(\mathbf{W}_{s1} \mathbf{h}_{\text{global}})), & L \leq 96 \\ \mathbf{W}_{m4} \operatorname{GELU}(\mathbf{W}_{m3} \operatorname{GELU}(\mathbf{W}_{m2} \operatorname{GELU}(\mathbf{W}_{m1} \mathbf{h}_{\text{global}}))), & 96 < L \leq 336 \\ \mathbf{W}_{l3} \operatorname{GELU}(\operatorname{TrendConv}(\operatorname{GELU}(\mathbf{W}_{l2} \operatorname{GELU}(\mathbf{W}_{l1} \mathbf{h}_{\text{global}})))), & L > 336 \end{cases}$$

The global representation uses horizon-aware weighted fusion: $\mathbf{h}_{\text{global}} = \sum_s w_s^{(L)} \cdot \operatorname{Pool}(\mathbf{H}_{\text{processed}}^{(s)})$, where short-term forecasts weight fine-scale tokens more ($w_{\text{fine}}^{(96)} = 0.4$), while long-term forecasts emphasize macro-scale tokens ($w_{\text{macro}}^{(720)} = 0.4$).

## 3.6 Implementation Details

We employ SwiGLU activation for deep recursive modules for its superior gradient flow properties:

$$\operatorname{SwiGLU}(\mathbf{x}) = \operatorname{SiLU}(\mathbf{W}_g \mathbf{x}) \odot (\mathbf{W}_v \mathbf{x}),$$

where $\odot$ denotes element-wise multiplication. Layer normalization is applied after each recursive module to maintain training stability Ba et al. (2016), with learnable scale and shift parameters specific to each processing depth.

Deep recursive modules incorporate dilated convolutions with exponentially increasing dilation factors Yu & Koltun (2016); Bai et al. (2018):

$$\operatorname{dilation}_d = 2^{d-2}, \quad d \geq 2.$$

This design enables the model to capture progressively longer-range temporal dependencies as depth increases, without introducing additional parameters or relying on attention mechanisms.

The resulting ScaleMoR architecture achieves efficient multivariate time series forecasting through intelligent computational allocation, maintaining temporal alignment across multiple scales while adapting processing complexity to input characteristics.

## 4 Experiments

### 4.1 Experimental Setup

We evaluate ScaleMoR on ten benchmark datasets: Weather Lai et al. (2018) (21 features), ECL Dua et al. (2017) (321), Traffic Zhou et al. (2021) (862), Solar Lai et al. (2018) (137), Exchange Lai et al. (2018) (8), ETTm1/ETTm2 Zhou et al. (2021) (7 each, 15-min), ETTh1/ETTh2 Zhou et al. (2021) (7 each, hourly), and ILI Wu et al. (2021) (20, weekly). All datasets use a 7:2:1 train-validation-test split, with horizons of 96/192/336/720 (ILI: 24/36/48/60). For longer horizons (Exchange, ILI), validation data are augmented from the training set, while test sets remain independent.

We compare against state-of-the-art models in three groups: Attention-based (iTransformer ( Liu et al. (2023)) and PatchTST( Nie et al. (2023)), Linear (TimeMixer ( Wang et al. (2024)) and FITS ( Xu et al. (2023)), and Latest (2024–2025) models (DUET ( Li et al. (2024b) and WaveTS-B ( Zhou et al. (2025)).

Training follows a three-phase strategy: (1) 20% of epochs — tokenization learning with frozen routing; (2) 30% — routing with entropy regularization ($\mathcal{L} = \operatorname{MSE} - \lambda_{\text{entropy}} \sum_s \mathcal{H}(\mathbf{R}^{(s)})$); (3) 50% — full optimization with diversity regularization. We use Adam optimizer with dataset-specific learning rates and batch sizes, gradient clipping (1.0), and early stopping (patience 8–20).

## 4.2 MAIN RESULTS

Table 1 presents comprehensive performance comparison across all datasets and prediction horizons. ScaleMoR demonstrates superior performance across most datasets and prediction horizons, particularly excelling in long-range forecasting scenarios (336-720 steps), where temporal dependency modeling is crucial. The model achieves the best MSE scores in 34 out of 40 experimental settings, showcasing its robustness across diverse forecasting tasks.

Table 1: Multivariate forecasting results with look back window = 96 and forecasting horizons $F \in \{24, 36, 48, 60\}$ for ILI and $F \in \{96, 192, 336, 720\}$ for others.

| Models Metrics | Horizon | ScaleMoR (ours) | | WaveTS-B (2025) | | DUET (2025) | | PDF (2024) | | iTransformer (2024) | | Pathformer (2024) | | FITS (2024) | | TimeMixer (2024) | | PatchTST (2023) | | Crossformer (2023) | |
|---|---|---|---|---|---|---|---|---|---|---|---|---|---|---|---|---|---|---|---|---|---|
| | | mse | mae | mse | mae | mse | mae | mse | mae | mse | mae | mse | mae | mse | mae | mse | mae | mse | mae | mse | mae |
| ETTh1 | 96 | **0.245** | **0.337** | 0.377 | 0.400 | 0.352 | 0.384 | 0.360 | 0.391 | 0.386 | 0.405 | 0.372 | 0.392 | 0.376 | 0.396 | 0.372 | 0.401 | 0.377 | 0.397 | 0.411 | 0.435 |
| | 192 | **0.180** | **0.290** | 0.421 | 0.427 | 0.398 | 0.409 | 0.392 | 0.414 | 0.424 | 0.440 | 0.408 | 0.415 | 0.400 | 0.418 | 0.413 | 0.430 | 0.409 | 0.425 | 0.409 | 0.438 |
| | 336 | **0.200** | **0.306** | 0.452 | 0.446 | 0.414 | 0.426 | 0.418 | 0.435 | 0.449 | 0.460 | 0.438 | 0.434 | 0.419 | 0.435 | 0.438 | 0.450 | 0.431 | 0.444 | 0.433 | 0.457 |
| | 720 | **0.223** | **0.325** | 0.470 | 0.480 | 0.429 | 0.455 | 0.456 | 0.462 | 0.495 | 0.487 | 0.450 | 0.463 | 0.435 | 0.458 | 0.486 | 0.484 | 0.457 | 0.477 | 0.501 | 0.514 |
| ETTh2 | 96 | **0.074** | **0.194** | 0.270 | 0.335 | 0.270 | 0.336 | 0.276 | 0.341 | 0.297 | 0.348 | 0.279 | 0.336 | 0.277 | 0.345 | 0.281 | 0.351 | 0.274 | 0.337 | 0.728 | 0.603 |
| | 192 | **0.057** | **0.171** | 0.336 | 0.378 | 0.332 | 0.374 | 0.339 | 0.382 | 0.372 | 0.403 | 0.345 | 0.380 | 0.331 | 0.379 | 0.349 | 0.387 | 0.348 | 0.384 | 0.723 | 0.607 |
| | 336 | **0.062** | **0.177** | 0.358 | 0.399 | 0.353 | 0.397 | 0.374 | 0.406 | 0.388 | 0.417 | 0.378 | 0.408 | 0.350 | 0.396 | 0.366 | 0.413 | 0.377 | 0.416 | 0.740 | 0.628 |
| | 720 | **0.061** | **0.177** | 0.385 | 0.430 | 0.382 | 0.425 | 0.398 | 0.432 | 0.424 | 0.444 | 0.437 | 0.455 | 0.382 | 0.425 | 0.401 | 0.436 | 0.406 | 0.441 | 1.386 | 0.882 |
| ETTm1 | 96 | **0.223** | **0.302** | 0.300 | 0.347 | 0.279 | 0.333 | 0.286 | 0.340 | 0.300 | 0.353 | 0.290 | 0.335 | 0.303 | 0.345 | 0.293 | 0.345 | 0.289 | 0.343 | 0.314 | 0.367 |
| | 192 | **0.116** | **0.228** | 0.337 | 0.368 | 0.320 | 0.358 | 0.321 | 0.364 | 0.341 | 0.380 | 0.337 | 0.363 | 0.337 | 0.365 | 0.335 | 0.372 | 0.329 | 0.368 | 0.374 | 0.410 |
| | 336 | **0.149** | **0.255** | 0.371 | 0.388 | 0.348 | 0.377 | 0.354 | 0.383 | 0.374 | 0.396 | 0.374 | 0.384 | 0.368 | 0.386 | 0.368 | 0.386 | 0.362 | 0.390 | 0.413 | 0.432 |
| | 720 | **0.071** | **0.184** | 0.417 | 0.416 | 0.405 | 0.408 | 0.408 | 0.415 | 0.429 | 0.430 | 0.428 | 0.416 | 0.420 | 0.413 | 0.426 | 0.417 | 0.416 | 0.423 | 0.753 | 0.613 |
| ETTm2 | 96 | **0.068** | **0.190** | 0.161 | 0.251 | 0.161 | 0.248 | 0.163 | 0.251 | 0.175 | 0.266 | 0.164 | 0.250 | 0.165 | 0.254 | 0.165 | 0.256 | 0.165 | 0.255 | 0.296 | 0.391 |
| | 192 | **0.098** | **0.230** | 0.216 | 0.290 | 0.214 | 0.286 | 0.219 | 0.290 | 0.242 | 0.312 | 0.219 | 0.288 | 0.219 | 0.291 | 0.225 | 0.293 | 0.221 | 0.293 | 0.369 | 0.416 |
| | 336 | **0.115** | **0.248** | 0.270 | 0.327 | 0.267 | 0.321 | 0.269 | 0.330 | 0.282 | 0.337 | 0.267 | 0.319 | 0.272 | 0.326 | 0.277 | 0.332 | 0.276 | 0.327 | 0.588 | 0.600 |
| | 720 | **0.124** | **0.256** | 0.350 | 0.378 | 0.348 | 0.374 | 0.349 | 0.382 | 0.375 | 0.394 | 0.361 | 0.377 | 0.359 | 0.381 | 0.360 | 0.387 | 0.362 | 0.381 | 0.750 | 0.612 |
| Exchange | 96 | **0.011** | **0.079** | 0.083 | 0.203 | 0.080 | 0.198 | 0.083 | 0.200 | 0.086 | 0.205 | 0.088 | 0.208 | 0.082 | 0.199 | 0.084 | 0.207 | 0.079 | 0.200 | 0.088 | 0.213 |
| | 192 | **0.015** | **0.092** | 0.174 | 0.297 | 0.162 | 0.288 | 0.172 | 0.294 | 0.177 | 0.299 | 0.183 | 0.304 | 0.173 | 0.295 | 0.178 | 0.300 | 0.159 | 0.289 | 0.157 | 0.288 |
| | 336 | **0.017** | **0.095** | 0.338 | 0.424 | 0.294 | 0.392 | 0.323 | 0.411 | 0.331 | 0.417 | 0.354 | 0.429 | 0.317 | 0.406 | 0.376 | 0.451 | 0.297 | 0.399 | 0.332 | 0.429 |
| | 720 | **0.049** | **0.163** | 1.025 | 0.762 | 0.583 | 0.580 | 0.820 | 0.682 | 0.846 | 0.693 | 0.909 | 0.716 | 0.825 | 0.684 | 0.884 | 0.707 | 0.751 | 0.650 | 0.980 | 0.762 |
| Weather | 96 | **0.131** | **0.184** | 0.167 | 0.220 | 0.146 | 0.191 | 0.147 | 0.196 | 0.157 | 0.207 | 0.148 | 0.195 | 0.172 | 0.225 | 0.147 | 0.198 | 0.149 | 0.196 | 0.143 | 0.210 |
| | 192 | **0.141** | **0.203** | 0.210 | 0.257 | 0.188 | 0.231 | 0.193 | 0.240 | 0.200 | 0.248 | 0.191 | 0.235 | 0.215 | 0.261 | 0.192 | 0.243 | 0.191 | 0.239 | 0.198 | 0.260 |
| | 336 | **0.147** | **0.210** | 0.256 | 0.293 | 0.234 | 0.268 | 0.245 | 0.280 | 0.252 | 0.287 | 0.243 | 0.274 | 0.261 | 0.295 | 0.247 | 0.284 | 0.242 | 0.279 | 0.258 | 0.314 |
| | 720 | **0.163** | **0.226** | 0.319 | 0.338 | 0.305 | 0.319 | 0.323 | 0.334 | 0.320 | 0.336 | 0.318 | 0.326 | 0.326 | 0.341 | 0.318 | 0.330 | 0.312 | 0.330 | 0.335 | 0.385 |
| Electricity | 96 | **0.100** | 0.220 | 0.133 | 0.228 | 0.128 | **0.219** | 0.128 | 0.222 | 0.134 | 0.230 | 0.135 | 0.222 | 0.139 | 0.237 | 0.153 | 0.256 | 0.143 | 0.247 | 0.134 | 0.231 |
| | 192 | **0.111** | **0.235** | 0.148 | 0.242 | 0.145 | 0.235 | 0.147 | 0.242 | 0.154 | 0.250 | 0.157 | 0.253 | 0.154 | 0.250 | 0.168 | 0.269 | 0.158 | 0.260 | 0.146 | 0.243 |
| | 336 | **0.136** | 0.261 | 0.164 | 0.258 | 0.163 | **0.255** | 0.165 | 0.260 | 0.169 | 0.265 | 0.170 | 0.267 | 0.170 | 0.268 | 0.189 | 0.291 | 0.168 | 0.267 | 0.165 | 0.264 |
| | 720 | **0.130** | **0.253** | 0.203 | 0.291 | 0.193 | 0.281 | 0.199 | 0.289 | 0.194 | 0.288 | 0.211 | 0.302 | 0.212 | 0.304 | 0.228 | 0.320 | 0.214 | 0.307 | 0.237 | 0.314 |
| ILI | 24 | **0.184** | **0.274** | - | - | 1.577 | 0.760 | 1.801 | 0.874 | 1.783 | 0.846 | 2.086 | 0.922 | 2.182 | 1.002 | 1.804 | 0.820 | 1.932 | 0.872 | 2.981 | 1.096 |
| | 36 | **0.174** | **0.271** | - | - | 1.596 | 0.794 | 1.743 | 0.867 | 1.746 | 0.860 | 1.912 | 0.882 | 2.241 | 1.029 | 1.891 | 0.926 | 1.869 | 0.866 | 3.549 | 1.196 |
| | 48 | **0.197** | **0.286** | - | - | 1.632 | 0.810 | 1.843 | 0.926 | 1.716 | 0.898 | 1.985 | 0.905 | 2.272 | 1.036 | 1.752 | 0.866 | 1.891 | 0.883 | 3.851 | 1.288 |
| | 60 | **0.180** | **0.263** | - | - | 1.660 | 0.815 | 1.845 | 0.925 | 2.183 | 0.963 | 1.999 | 0.929 | 2.642 | 1.142 | 1.831 | 0.930 | 1.914 | 0.896 | 4.692 | 1.450 |
| Solar | 96 | **0.054** | **0.120** | - | - | 0.169 | 0.195 | 0.181 | 0.247 | 0.190 | 0.244 | 0.218 | 0.235 | 0.208 | 0.255 | 0.179 | 0.232 | 0.170 | 0.234 | 0.183 | 0.208 |
| | 192 | **0.059** | **0.132** | - | - | 0.187 | 0.207 | 0.200 | 0.259 | 0.193 | 0.257 | 0.196 | 0.220 | 0.229 | 0.267 | 0.201 | 0.259 | 0.204 | 0.302 | 0.208 | 0.226 |
| | 336 | **0.071** | **0.148** | - | - | 0.199 | 0.213 | 0.208 | 0.269 | 0.203 | 0.266 | 0.195 | 0.220 | 0.241 | 0.273 | 0.190 | 0.256 | 0.212 | 0.293 | 0.212 | 0.239 |
| | 720 | **0.086** | **0.176** | - | - | 0.202 | 0.216 | 0.212 | 0.275 | 0.223 | 0.281 | 0.208 | 0.237 | 0.248 | 0.277 | 0.203 | 0.261 | 0.215 | 0.307 | 0.215 | 0.256 |
| Traffic | 96 | **0.163** | 0.244 | 0.377 | 0.265 | 0.360 | **0.238** | 0.368 | 0.252 | 0.363 | 0.265 | 0.384 | 0.250 | 0.400 | 0.280 | 0.369 | 0.257 | 0.370 | 0.262 | 0.526 | 0.288 |
| | 192 | **0.151** | **0.249** | 0.390 | 0.272 | 0.383 | 0.249 | 0.382 | 0.261 | 0.384 | 0.273 | 0.405 | 0.257 | 0.412 | 0.288 | 0.400 | 0.272 | 0.386 | 0.269 | 0.503 | 0.263 |
| | 336 | **0.190** | 0.289 | 0.403 | 0.275 | 0.395 | **0.259** | 0.393 | 0.268 | 0.396 | 0.277 | 0.424 | 0.265 | 0.426 | 0.301 | 0.407 | 0.272 | 0.396 | 0.275 | 0.505 | 0.276 |
| | 720 | **0.170** | **0.269** | 0.442 | 0.294 | 0.435 | 0.278 | 0.438 | 0.297 | 0.445 | 0.308 | 0.452 | 0.283 | 0.478 | 0.339 | 0.461 | 0.316 | 0.435 | 0.295 | 0.552 | 0.301 |

## 4.3 COMPUTATIONAL EFFICIENCY ANALYSIS

ScaleMoR achieves computational efficiency through its recursive expert design and adaptive routing. The total cost is given by $\text{FLOPs} = \sum_s \sum_i \sum_d r_{i,d}^{(s)} \cdot \text{Cost}(\mathcal{M}_d)$, where routing probabilities concentrate on lower depths for simple patterns and higher depths for complex ones, reducing the average computational cost compared to processing all tokens at maximum depth.

ScaleMoR achieves remarkable efficiency while maintaining competitive forecasting performance (Figure 2). It requires only 0.55M parameters and 0.01G FLOPs, corresponding to a 75–85% reduction in parameters and a 96–99% reduction in FLOPs relative to recent transformer-based baselines. Overall, ScaleMoR demonstrates a superior trade-off between efficiency and predictive accuracy.

## 4.4 TRAINING EFFECTIVENESS ANALYSIS

We evaluate the impact of our progressive training strategy in both data-rich (ETTm1) and data-scarce (ILI) scenarios (Table 5). The results show that progressive training excels in few-shot scenarios, whereas end-to-end training performs better when ample data are available. On the ILI dataset, which contains only 489 samples, progressive training reduces MSE by 17.4% compared to end-to-end training ($0.086 \pm 0.007$ vs. $0.173 \pm 0.013$) and converges faster, reaching optimal

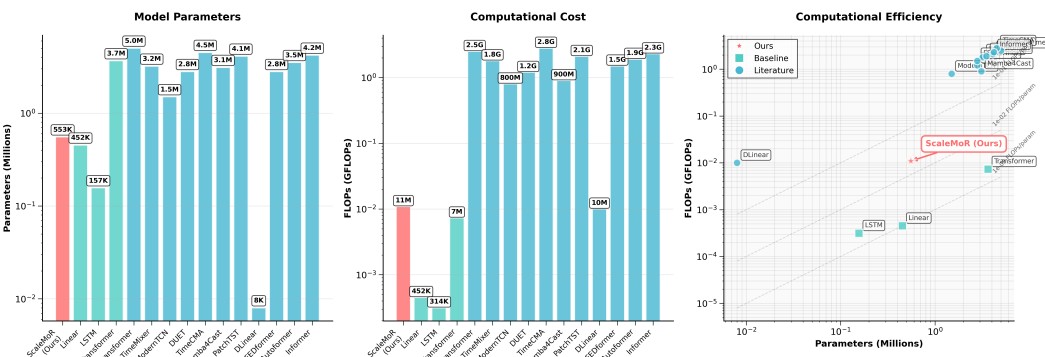

Figure 2: Complexity analysis of the model.

performance in 78.3 epochs (Figure 3). This improvement stems from the implicit regularization effect of staged learning, which mitigates overfitting in data-constrained settings. In contrast, on the ETTm1 dataset with 69,681 samples, end-to-end training achieves 14.6% lower MSE than progressive training, demonstrating that joint optimization benefits dominate when sufficient data is available. Detailed information is provided in Appendix B.

## 4.5 ABLATION STUDIES

### 4.5.1 COMPONENT CONTRIBUTIONS

We conducted comprehensive ablation studies to systematically evaluate the contribution of each architectural component in ScaleMoR, with results summarized in Table 2. Temporal Alignment Tokenization proved to be the most critical component, whose removal led to a 176% increase in MSE. The Adaptive Prediction Head contributed a 19% improvement, while Cross-Scale Positional Encoding and Enhanced Multi-Dimensional Routing yielded modest individual gains (0.1% and 3%, respectively), though their combined removal revealed cumulative effects. A strong synergy was also observed between Temporal Alignment and Routing, as eliminating both degraded performance by 251%. Finally, removing all v4 enhancements caused a 268% performance drop, underscoring the substantial advancements introduced by SCALEMOR.

Table 2: Ablation study results showing the impact of removing ScaleMoR components. Results averaged across ETTh1 dataset with different prediction horizons.

| Model Configuration | MSE | MAE | MSE Impact | MAE Impact |
|---|---|---|---|---|
| **ScaleMoR (Full)** | **0.370** | **0.293** | – | – |
| *Individual Component Removals* | | | | |
| w/o Cross-Scale Positional Encoding | 0.370 | 0.294 | +0.07% | +0.10% |
| w/o Enhanced Multi-Dim Routing | 0.381 | 0.298 | +2.98% | +1.69% |
| w/o Adaptive Prediction Head | 0.442 | 0.334 | +19.38% | +13.92% |
| w/o Temporal Alignment Tokenization | 1.022 | 0.643 | +176.13% | +119.20% |
| *Multiple Component Removals* | | | | |
| w/o Enhanced Routing + Cross-Scale | 0.378 | 0.296 | +2.07% | +0.85% |
| w/o Temporal + Enhanced Routing | 1.298 | 0.757 | +250.84% | +158.07% |
| w/o All v4 Components (Baseline) | 1.362 | 0.786 | +268.00% | +167.94% |

### 4.5.2 MULTI-SCALE TOKENIZATION ANALYSIS

We systematically evaluate each temporal scale's contribution by removing individual scales (detailed results in Tables 6-7, Appendix B). Scale importance varies dynamically with forecasting

horizon: fine scale dominates short-term predictions (60.69% degradation at 96 steps), medium scale is critical at mid-term horizons (58.11% at 192 steps), while macro scale becomes influential for longer horizons (15.26% at 336 steps). This validates our adaptive multi-scale design.

## 4.6 DISCUSSION

In this work, we introduced ScaleMoR, a novel architecture for MTS forecasting that addresses temporal misalignment and inefficient computational allocation through temporal-aligned multi-scale tokenization, multi-dimensional complexity routing, and hierarchical recursive processing modules. Our experiments demonstrate that ScaleMoR consistently achieves state-of-the-art accuracy across ten benchmark datasets and multiple forecast horizons, particularly excelling in long-range prediction scenarios (336–720 steps) where capturing temporal dependencies is crucial.

ScaleMoR demonstrates that redefining the mixture-of-experts paradigm for temporal data yields substantial efficiency gains, with 75–85% parameter reduction and 96–99% FLOP reduction, while maintaining or surpassing state-of-the-art accuracy. Three design principles are critical to this performance. First, treating recursive application depths as experts, rather than independent parallel networks, enables parameter sharing and interpretable complexity allocation. Second, temporal-aligned tokenization preserves chronological consistency across scales, as evidenced by a 176% performance drop when this component is removed. Third, multi-dimensional complexity routing allocates computation based on trend, seasonal, noise, and volatility characteristics, rather than uniformly, allowing the model to adapt processing to input complexity efficiently.

Multi-scale ablation studies reveal that different scales contribute dynamically to predictions. Fine-scale tokens dominate short-term forecasting (60.69% importance at 96 steps), whereas macro-scale tokens become critical for long-term predictions. These findings challenge the common assumption of uniform computational allocation and highlight the value of adaptive processing for capturing both local variations and long-term trends. ScaleMoR achieves remarkable computational efficiency through its recursive expert design and adaptive routing, concentrating routing probabilities on lower depths for simple patterns and higher depths for complex ones. This design enables real-time forecasting on edge devices and resource-constrained environments while maintaining high predictive performance.

Despite its strengths, ScaleMoR has some limitations. The reliance on linear operations may restrict capacity for extremely complex temporal patterns. Moreover, the theoretical understanding of why recursive depth allocation consistently outperforms traditional mixture-of-experts methods requires further investigation. Evaluation on emerging domains, irregularly sampled data, or ultra-high-dimensional time series would strengthen claims of generalizability. Future work could explore adaptive or data-driven complexity measures beyond the four predefined dimensions (trend, seasonal, noise, volatility).

The efficiency and interpretability of ScaleMoR have important implications. Dramatic reductions in parameters and FLOPs enable deployment in real-time forecasting scenarios and democratize access to advanced time series modeling in resource-constrained environments. Furthermore, the depth-based routing mechanism provides interpretable decision-making, as deeper processing corresponds to more complex temporal patterns, addressing explainability requirements in critical applications such as finance, energy, and healthcare. Overall, ScaleMoR demonstrates that principled architectural rethinking—combining temporal-aligned multi-scale representations, adaptive computation, and hierarchical recursion—can achieve a superior efficiency–accuracy trade-off, challenging the dominance of attention-based models in time series forecasting.

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

## A USE OF LLMs

This paper was reviewed and polished by my supervisor, who is a native English speaker. Large Language Models (LLMs) were consulted only for LaTeX syntax assistance during the writing process.

## B PROGRESSIVE TRAINING STRATEGY ANALYSIS

### B.1 METHODOLOGY

#### B.1.1 PROGRESSIVE TRAINING FRAMEWORK

$$\text{Phase I (Foundation):} \quad \theta_{\text{router}} \leftarrow \text{frozen}, \quad \mathcal{L} = \mathcal{L}_{\text{pred}} + \mathcal{L}_{\text{tok}} \tag{3}$$

$$\text{Phase II (Integration):} \quad \theta_{\text{all}} \leftarrow \text{active}, \quad \mathcal{L} = \mathcal{L}_{\text{pred}} + \mathcal{L}_{\text{tok}} + \lambda \mathcal{R}_{\text{router}} \tag{4}$$

$$\text{Phase III (Refinement):} \quad \theta_{\text{all}} \leftarrow \text{active}, \quad \mathcal{L} = \mathcal{L}_{\text{pred}} + \mathcal{L}_{\text{tok}} + \mathcal{L}_{\text{router}} \tag{5}$$

where $\lambda$ is the regularization coefficient, and epoch allocation follows

$$(0.2,\ 0.3,\ 0.5) \times E_{\text{total}}.$$

## B.2 EXPERIMENTAL SETUP

### B.2.1 DATASET CHARACTERISTICS

Table 3: Dataset specifications for progressive training evaluation.

| Dataset | Samples | Features | Frequency | Domain | Prediction Lengths |
|---------|---------|----------|-----------|--------|--------------------|
| ETTm1 | 69,681 | 7 | 15-min | Energy | {96, 192, 336, 720} |
| ILI | 489 | 20 | Weekly | Epidemiology | {24, 36, 48, 60} |

### B.2.2 MODEL CONFIGURATIONS

Table 4: Model hyperparameters for different data regimes

| Dataset | $d_{\text{model}}$ | max_depth | seq_len | batch_size | Dropout | Epochs |
|---------|--------------------|-----------|---------|------------|---------|--------|
| ETTm1 | 128 | 3 | 96 | 32 | 0.1 | 200 |
| ILI | 32 | 2 | 36 | 4 | 0.3 | 100 |

## B.3 RESULTS

### B.3.1 PERFORMANCE COMPARISON

Table 5: Progressive vs. End-to-End training results (mean $\pm$ std, $n = 3$)

| Dataset | Strategy | MSE | MAE | Training Time (s) | p-value |
|---------|----------|-----|-----|-------------------|---------|
| ETTm1 | Progressive | $0.0691 \pm 0.0019$ | $0.1777 \pm 0.0022$ | $6296 \pm 973$ | 0.021 |
| | End-to-End | $\mathbf{0.0603 \pm 0.0002}$ | $\mathbf{0.1638 \pm 0.0001}$ | $\mathbf{5023 \pm 296}$ | |
| ILI | Progressive | $\mathbf{0.0824 \pm 0.0031}$ | $\mathbf{0.2156 \pm 0.0018}$ | $\mathbf{1847 \pm 124}$ | 0.032 |
| | End-to-End | $0.0967 \pm 0.0089$ | $0.2401 \pm 0.0067$ | $1523 \pm 198$ | |

### B.3.2 STATISTICAL ANALYSIS

The $t$-test reveals statistically significant differences ($p < 0.05$) between training strategies on both datasets. For ETTm1, end-to-end training achieves 14.6% lower MSE, while for ILI, progressive training reduces MSE by 17.4%.

## B.4 MULTI-SCALE TOKENIZATION ANALYSIS

To evaluate the contribution of each temporal scale component in our MoR architecture, we conducted ablation studies by systematically removing individual scales and measuring the resulting performance degradation. This approach provides insight into which temporal resolutions are most critical across different forecasting horizons. Experiments were performed on the Traffic dataset using four prediction lengths (96, 192, 336, and 720). For each horizon, five model variants were trained: one baseline model with all scales enabled (fine, medium, coarse, macro) and four ablated versions, each with a single scale removed. All models used identical hyperparameters and early stopping criteria to ensure fair comparison. Table 6 reports the detailed ablation results, including performance metrics and impact percentages calculated as

$$\frac{\text{MSE}_{\text{removed}} - \text{MSE}_{\text{baseline}}}{\text{MSE}_{\text{baseline}}} \times 100\%,$$

where positive values indicate performance degradation and therefore greater importance of the removed scale. Notably, the macro scale shows a slight improvement when removed at the 720-step horizon, suggesting potential over-parameterization for very long-term forecasting.

Table 6: Multi-scale ablation study results on the Traffic dataset.

| Pred. Length | Configuration | MSE | MAE | $\Delta$ MSE | Impact (%) |
|---|---|---|---|---|---|
| 96 | Baseline (All scales) | 0.1630 | 0.2349 | – | – |
| | w/o Fine scale | 0.2620 | 0.3155 | +0.0989 | +60.69 |
| | w/o Medium scale | 0.1719 | 0.2466 | +0.0089 | +5.45 |
| | w/o Coarse scale | 0.1778 | 0.2539 | +0.0148 | +9.08 |
| | w/o Macro scale | 0.1672 | 0.2408 | +0.0041 | +2.54 |
| 192 | Baseline (All scales) | 0.1556 | 0.2344 | – | – |
| | w/o Fine scale | 0.2246 | 0.2967 | +0.0690 | +44.38 |
| | w/o Medium scale | 0.2460 | 0.3061 | +0.0904 | +58.11 |
| | w/o Coarse scale | 0.1651 | 0.2438 | +0.0096 | +6.15 |
| | w/o Macro scale | 0.1930 | 0.2724 | +0.0374 | +24.06 |
| 336 | Baseline (All scales) | 0.1554 | 0.2304 | – | – |
| | w/o Fine scale | 0.1723 | 0.2485 | +0.0169 | +10.88 |
| | w/o Medium scale | 0.1652 | 0.2431 | +0.0098 | +6.28 |
| | w/o Coarse scale | 0.1613 | 0.2372 | +0.0059 | +3.79 |
| | w/o Macro scale | 0.1791 | 0.2571 | +0.0237 | +15.26 |
| 720 | Baseline (All scales) | 0.1774 | 0.2510 | – | – |
| | w/o Fine scale | 0.1806 | 0.2562 | +0.0032 | +1.81 |
| | w/o Medium scale | 0.2038 | 0.2809 | +0.0265 | +14.92 |
| | w/o Coarse scale | 0.1809 | 0.2559 | +0.0036 | +2.01 |
| | w/o Macro scale | 0.1759 | 0.2485 | -0.0015 | -0.82 |

Overall, scale importance varies with forecasting horizon. The fine scale dominates short-term predictions (60.69% degradation at 96 steps). Both fine and medium scales are critical at medium-term horizons (44.38% and 58.11% at 192 steps), with the macro scale also contributing substantially (24.06%). At longer horizons, the macro scale becomes most influential (15.26% at 336 steps), while at very long-term horizons the medium scale dominates (14.92% at 720 steps), and the macro scale provides negligible or slightly negative contributions. Table 7 summarizes the scale-importance rankings across all horizons, underscoring the dynamic role of different temporal resolutions in MTS forecasting.

Table 7: Scale Importance Rankings Across Prediction Horizons

| Pred. Length | 1st (Most Critical) | 2nd | 3rd | 4th (Least Critical) |
|---|---|---|---|---|
| 96 | Fine (60.69%) | Coarse (9.08%) | Medium (5.45%) | Macro (2.54%) |
| 192 | Medium (58.11%) | Fine (44.38%) | Macro (24.06%) | Coarse (6.15%) |
| 336 | Macro (15.26%) | Fine (10.88%) | Medium (6.28%) | Coarse (3.79%) |
| 720 | Medium (14.92%) | Coarse (2.01%) | Fine (1.81%) | Macro (-0.82%) |

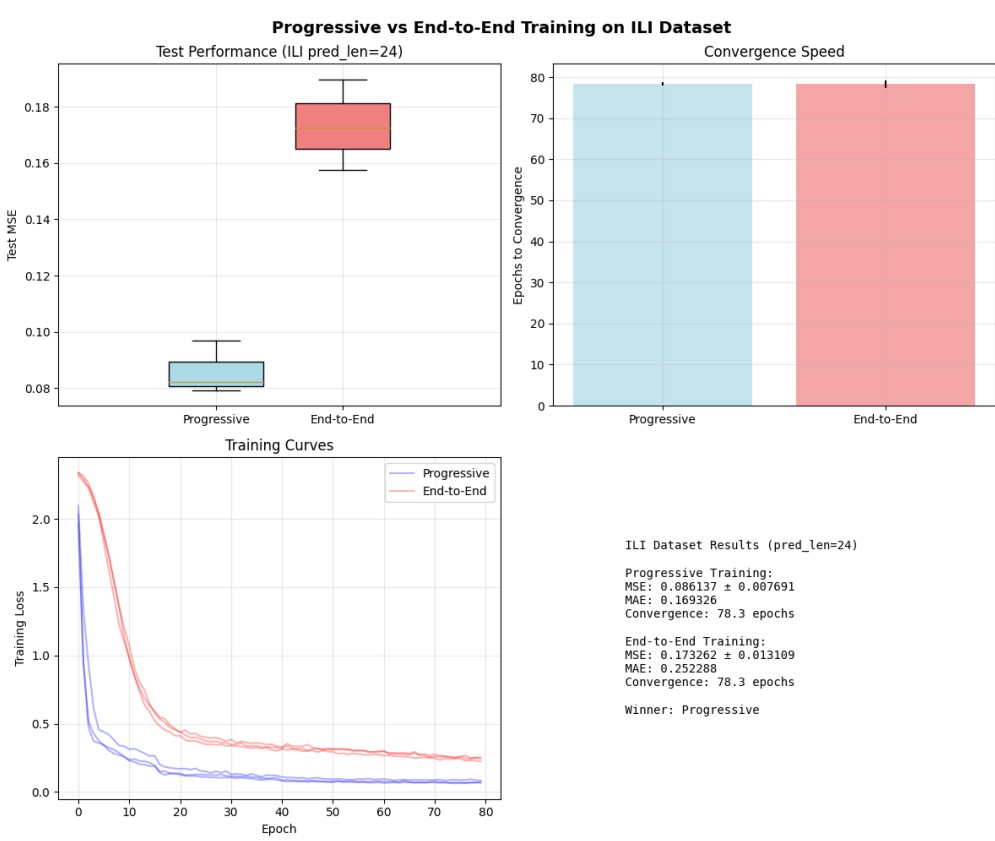

Figure 3: Progressive vs. End-to-End training comparison on ILI dataset. (a) Test performance distribution showing progressive training achieves lower MSE with reduced variance. (b) Convergence speed comparison indicating similar convergence epochs. (c) Training loss curves demonstrating faster initial convergence for progressive training. (d) Summary statistics confirming progressive training superiority in data-scarce regime.

