# OpenReview forum: "ScaleMoR: Multi-Scale Mixture of Recursive Linear Experts for Time Series Forecasting"
_ICLR.cc/2026/Conference — ICLR 2026 Conference Withdrawn Submission_

### Official Review · Reviewer_wH4T · 2025-10-27

**Soundness:** 3
**Presentation:** 1
**Contribution:** 2
**Rating:** 2
**Confidence:** 4

**Summary:**

The paper proposes ScaleMoR, a multiscale refinement framework for multivariate time-series (MTS) forecasting that integrates a Gaussian-weighted interpolation module to mitigate temporal misalignment and recursively improves coarse forecasts at finer scales. The method aims to balance accuracy and efficiency, reporting solid results on a broad suite of datasets (cf. Table 1). While the engineering is thoughtful, the narrative remains largely heuristic: it does not articulate why the design should improve predictability under non-stationarity, nor provide analyses that tie components to concrete failure modes of MTS forecasting. Clearer theoretical/diagnostic grounding would elevate the contribution.

**Strengths:**

- Methodological practicality. The Gaussian-weighted interpolation serves as the most critical component, addressing temporal misalignment between different scales.

- Experiments (including appendices) cover many datasets and settings, showing reasonable generalization power.

- The method achieves a good accuracy–efficiency trade-off; Table 1 shows competitive results across several horizons/datasets.

**Weaknesses:**

- Heuristic framing. The work does not deeply engage with core MTS issues (predictability under non-stationarity, error compounding, spectral/nonlinear regime shifts). Explanations for why ScaleMoR helps in these regimes are not diagnostic or theoretical.

- Many problems in presentation: equations after L226 are unnumbered; Figure 1 has overfull labels and mis-typeset math; TrendConv is undefined (L276); duplicated paragraphs at L70/L77.

- Despite substantial exposition, Multi-Dim Routing yields only 1.69% MAE improvement. This could even be noise. Missing WaveTS-B on ILI results are unexplained.

- The Multi-dimensional Complexity Routing mechanism does not specify how sample distributions are desired across depths, leaving potential bias/imbalance unaddressed.

**Questions:**

How are samples distributed across depths in the Multi-dimensional Complexity Routing module? Is the depth assignment correlated with the input scale choices, and does this routing demonstrably improve predictability (with metrics/diagnostics beyond MAE/MSE)?

---

### Official Review · Reviewer_qTWi · 2025-10-29

**Soundness:** 2
**Presentation:** 1
**Contribution:** 3
**Rating:** 2
**Confidence:** 3

**Summary:**

This paper proposes ScaleMoR, a multi-scale mixture of experts approach for long-term time series forecasting. The method claims to address misalignment issues in multi-scale processing through Gaussian interpolation and employs a wavelet-based mixture of experts framework. The architecture decomposes time series into trend, seasonal, noise, and volatility components, and processes them through a hierarchical recursive module with SwiGLU activations instead of traditional transformer attention mechanisms. The authors evaluate their method on standard long-term forecasting benchmarks, comparing primarily against transformer-based baselines, and report strong empirical results with prediction horizons of at least 96 steps.

**Strengths:**

Strong empirical results: The experimental results look very good across the evaluated benchmarks, demonstrating competitive performance for long-term forecasting tasks.

Computational efficiency consideration: The paper addresses the important issue of uniform allocation of computational resources through a mixture of experts framework, which is a valid concern for practical applications.

**Weaknesses:**

## Weaknesses

1. **Unclear main contribution**: The abstract and introduction introduce many ideas without clearly articulating the single most significant contribution. It's difficult to understand what the key innovation is—whether it's the alignment mechanism, the feature decomposition, the adaptive computation, or their combination—and how these components complement each other. After reading the paper there are more questions about the contribution and the choice of benchmark (see later questions part)

2. **Misalignment problem insufficiently motivated**: The claimed misalignment problem in multi-scale processing is not clearly explained. The reviewer can only conceptualize this as an issue in online settings, but this is not clarified. The intuition for how Gaussian interpolation solves this problem is not adequately developed.

3. **Poor correspondence between text and figures**: Figure 1 lacks sufficient legends and does not clearly correspond to the method sections. Critical components mentioned in the text (depth router, 4D trend analysis, SwiGLU recursive modules) are difficult to locate in the figure. This makes the paper very difficult to follow.

4. **Confusing and inconsistent notation**: The mathematical notation is unclear and inconsistent. For example:
   - The relationship between small t and capital T is ambiguous (lines 196-199)
   - Subscripts are omitted inconsistently (line 202)
   - Dimensionality information is missing throughout
   - The meaning of various indices (i, 1 in W₁) is unclear

5. **Questionable benchmark choices**: The paper does not use transformer layers or attention mechanisms (at least that's my impression after reading, see questions part below), yet compares primarily against transformer-based methods. If the architecture is fundamentally different, comparisons with other non-transformer methods (e.g., tree-based models for tabular data) would be more appropriate and informative.

6. **Adaptive computation mechanism unclear**: A key claimed contribution is adaptive selection of complexity and computational resources, but where and how this is achieved in the architecture is never clearly explained. This is mentioned in Figure 1 but not adequately described in the method section. (I would love to know!)

7. **Role of wavelets unclear**: Wavelets are mentioned as a key component (wavelet-based mixture of experts), but their actual role and implementation in the method is never clearly described. It's unclear if wavelets are even used in the final implementation. (maybe I read it wrong and it is only for background?)

8. **Inconsistent scale definitions**: The paper introduces a "large time window" of 24 steps as representative of long-term patterns, but then evaluates on horizons of 96+ steps, which is the current standard. The choice of scales (2, 6, 12, 24) is not justified.

9. **Confusing technical details**: Important details are confusing, such as:
   - How does the hierarchical recursive processing module actually process temporal information without state?
   - Does Gaussian interpolation introduce supersampling and additional computational cost?
   - Why limit decomposition to exactly four components (trend, seasonal, noise, volatility)?

10. **Architecture description unclear**: Even after a second reading, it remains unclear how information flows through the architecture, how the different components connect, and what the actual computational graph looks like.

## Minor Issues

- Typo: "volatility" is misspelled in the figure
- The related work section might be better positioned later in the paper, closer to the discussion

## Recommendation

The paper shows promising empirical results, but the presentation needs major revision before it can be properly evaluated. The main contribution must be clearly articulated, the method description must be substantially clarified with better notation and figure-text correspondence, and the role of key components (wavelets, adaptive computation) must be explicitly described.

**Questions:**

I have listed structured questions (with help of LLM) in the above weakness part. I am going to say here my honest thoughts when reading the paper as it presents, and hopefully this can help you understand how a new reader perceives your paper. Hopefully you can have a sense of my raw feelings when reading the paper. These questions are genuine and I can increase my score if you can help in clarifying them.



Review: 1786_ScaleMoR_Multi_Scale_Mixture of Experts
Abstract
Reading the abstract, my first comment is that a large time window of 24 steps seems like a macro window, but for current long-term time series forecasting standards, we are predicting at least 96 steps into the future. Secondly, the abstract introduces many ideas without clearly highlighting what the single most significant contribution is, or what the key trick is that makes the method work. If all components work equally well, I fail to see the consistent theme or how they complement each other.

Introduction
Line 46: The quadratic cost of transformers with softmax attention is indeed an issue, but linear transformers achieve linear cost. There are ways to remedy this, though they are not mentioned here.

Line 55 and following paragraphs: The multiscale kernel processing has been introduced before. After line 55, the authors acknowledge that multiscale approaches have been used previously, and their criticism focuses on alignment issues. However, the authors are proposing a wavelet-based mixture of experts approach, and I don't see how the proposed method intuitively addresses the misalignment of multi-scale processing.

Line 60: The point about uniform allocation of computational resources is valid. Mixture of experts has been used widely in time series forecasting. What appears new to this reviewer is the wavelet-based mixture of experts approach to capture multi-scale dynamics, which addresses sequential dependency and efficient computation. This makes sense conceptually.

Line 74: The mixture of experts approach here has a nice property of sharing linear transformations. I think of it as applying different transformations that serve as different experts.

Line 84: It's still unclear where misalignment occurs. At any given time point, the coarse scale prediction will be for a different scale and lookback window when predicting into the future. Is this misalignment only an issue in an online setting where new data points arrive and you need to update the prediction for the next 96-step window immediately? The weighted interpolation mechanism that produces aligned tokens at four scales is mentioned, but at the introduction stage, I don't understand what it is. I'm very confused and need to read this later.

I'm skipping the related work section for now. After reading the introduction, I have so many questions about the proposed work that I don't want to get distracted. Perhaps the related work should be moved to later sections, closer to the discussion.

Method Section
Line 157 and Figure 1: Things begin to make more sense here. I would recommend adding more legends to Figure 1 so that when reading it, I can understand what I'm looking at.

Line 190: I still fail to see how this suffers from misalignment. The only way I can conceptualize it is in an online setting, so this needs clarification.

Scale choice: Why the choice of 2, 6, 12, and 24 for the four scales?

Gaussian interpolation: I grasp the general idea that it attempts to interpolate and align time steps exactly, but would this include generating supersampling and thereby increase computational cost?

Line 199: I'm confused about the notation for small t. Is it related to a particular instance in your capital T?

Line 196: What is the i notation, and what is the total dimensionality here?

It now appears that small t is an iterator for W, which is an interpolation matrix. I'm confused about the dimensions—providing explicit dimensions would help. You're obviously omitting the i subscript in the line 202 equation. What does the subscript 1 in W₁ refer to compared to line 199?

Equation 1, Line 215: It's interesting that you decompose into trend, seasonal, noise, and volatility. This is helpful, as I've seen similar work that extracts fundamental characteristics. However, why limit to these four components? Other work I've seen leaves flexibility to incorporate additional learned features from the time series.

Section 3.3: Hierarchical Recursive Processing Module
This section doesn't actually use a transformer. It appears to be using a recursive processing module—is this RNN-type or feed-forward? It's described as hierarchical recursive processing without state, so it's not an RNN. It replaces the attention mechanism but doesn't seem to have state information. How is time being processed? I'm still confused. I don't quite understand how information flows forward. I'm also trying to reference back to Figure 1, but I can't locate this component clearly. The linear-layer norm-GLU-dropout-linear-layer norm seems to be the input section. Is this replacing the MLP in the transformer to enrich features?

Where is the depth router in Figure 1? Figure 1 clearly indicates there's a depth router for adaptive selection based on complexity, but the 4D trend analysis jumps directly to the recursive linear component.

Section 3.4: Cross-Scale Position Encoding
Relative position encoding is a standard idea. Does this correspond to the middle graph in Figure 1? However, Figure 1 shows a 3GLU recursive component that I don't see described in Section 3.4. The sections and figures don't correspond well—the figures need legends.

The position embedding appears to go directly into the yellow box inside the green background, then to the prediction head. Where is this discussed?

Section 3.6: Swish GLU and Activation
I see different depths mentioned here, along with implementation details about activation. This is for gradient flow properties—that's acceptable.

At this point, I've finished reading Section 3 in detail.

Results
Your results look very good, although I'm still confused about the method. I don't understand where the adaptive selection of complexity and computation is achieved. Where is this written?

Also, your results show prediction horizons of at least 96 steps. So what is the 24 you introduced earlier? Why is 24 considered a long-term pattern?

It seems to me that you're not actually using an attention mechanism or any transformer layers in the architecture. So why are you comparing yourself primarily with transformer architectures?

Overall Assessment and Understanding
After a second reading, I now understand the figure better. The recursive component describes how you process with layer normalization, and you expand one of the SwiGLU blocks. The structure is about different depths of stacking these components. After input, you have a recursive layer for processing, then you stack several SwiGLU layers together with skip connections from each depth of the SwiGLU. The SwiGLU uses GLU and element-wise manipulation—is that attention or not?

My main takeaway is: if you preprocess your data, make it aligned, and extract seasonality, noise, and volatility (note: there's a typo—"volatility" is misspelled in your graph), and combine them together, you don't need attention or transformers, yet the output predictions are very good. This might be true, but I'm confused about the contribution. If you're not using a transformer, why is your benchmark transformers? Shouldn't you compare with tree-based models? We know that for well-organized tabular data, tree-based models also perform well.

I'm still confused about your main contribution. Earlier, you mentioned wavelets somewhere—where are wavelets actually used? Are they even used?

Key Unresolved Questions
How is adaptive computational resource allocation achieved? This is very interesting but unclear.
What exactly is the misalignment problem and how does your method solve it?
Why compare against transformer baselines if you're not using transformers?
What is the role of wavelets in the actual implementation?
Why these specific scale choices (2, 6, 12, 24)?

---

### Official Review · Reviewer_bJ4i · 2025-11-01

**Soundness:** 1
**Presentation:** 2
**Contribution:** 2
**Rating:** 2
**Confidence:** 3

**Summary:**

The paper introduces ScaleMoR for multivariate time-series forecasting. Instead of traditional Mixture-of-Experts (MoE) with multiple independent experts, it treats recursive depths of shared linear layers as experts. It includes 1) Temporal-aligned multi-scale tokenization: keeps time consistency when combining short- and long-term patterns using Gaussian interpolation. 2) Multi-dimensional routing: decides how much computation each token gets based on trend, seasonality, noise, and volatility. 3) Recursive modules: replace attention with lightweight linear and convolutional layers, mixing depth and efficiency. 4) Adaptive prediction head: changes structure depending on the forecast horizon.

**Strengths:**

1. ScaleMoR introduces a structure by redefining the mixture-of-experts framework through recursive depth sharing, turning computation depth into an interpretable expert dimension. The model is clearly designed and is practical efficiency

2. The structure has a temporally aligned multi-scale tokenization scheme, semantically meaningful multi-dimensional routing, and lightweight recursive modules that remove the need for attention.

3. The paper is easy to follow. The approach demonstrates that conditional computation can be both interpretable and highly efficient, offering a good direction for real-time and resource-constrained time-series forecasting.

**Weaknesses:**

1. The reported experimental results appear to have serious issues. The outcomes in Table 1 look unusually high and conflict with other parts of the paper.

First, the improvements over existing baselines are unrealistically large, suggesting possible data leakage. For instance, in the Exchange dataset, the time series roughly follows an AR(1) process, for which a simple naïve repeat predictor is nearly optimal. Under the same setup, the naïve predictor achieves an MSE of about 0.078, close to other baselines, yet ScaleMoR reports 0.011, which far exceeds both the naïve baseline and all other models. This implies the model might have unintentionally accessed future information (e.g., future statistics). To rule out data leakage, I suggest training ScaleMoR on several synthetic random walk series and comparing its performance to the naïve predictor. If ScaleMoR performs similarly or worse, it would confirm that it does not exploit unavailable future noise information.

Second, several statements about Table 1 conflict with the numbers themselves. For example, line 329 claims that “the model achieves the best MSE scores in 34 out of 40 settings,” yet Table 1 shows ScaleMoR outperforming all baselines in every case. Likewise, the ETTh1 full-model MSE in Table 2 differs greatly from that in Table 1, being much closer to prior baselines. This inconsistency suggests there may have been merged or mislabeled experiment settings. Since Table 6 still resembles the questionable Table 1 results, the authors should provide a clear, systematic explanation of their experimental implementation and data handling.

2. Some technical descriptions in the paper are misleading. The text repeatedly states that the model "only depends on linear operations," but the use of GELU and SwiGLU introduces clear nonlinearity. The authors may be confusing "linear operations" with "linear layers," yet a network containing nonlinear activations cannot be considered purely linear, only if all computations from input to output can be expressed as a single linear transformation would this be accurate. Given that the Hierarchical MoR and the prediction head effectively stack multiple MLP layers, the model has strong nonlinear expressivity. The authors should revise such claims, perhaps describing the method as having linear complexity with respect to sequence length, rather than being "fully linear."

3. Several architectural choices seem overly heuristic and lack strong theoretical motivation. For example, the design of prediction head changes drastically depending on the forecast length L, with 96 and 336 acting as arbitrary cutoffs. Each range uses different MLP depths, activation functions, and even introduces a TrendConv layer for the longest horizon. Such heavily tuned design choices may appear to overfit specific benchmarks rather than generalize. To demonstrate robustness, the authors should either adopt a consistent architecture across horizons or allow the structure to be data-driven and learned, which would make the evaluation results more trustworthy.

**Questions:**

1. Could the authors provide more detailed experimental settings and hyperparameters to strengthen the credibility of their main results?

2. Releasing an anonymized codebase with complete experimental configurations would greatly improve transparency and reproducibility.

3. The layout of Figure 1 could be further refined. It currently occupies too much space, the font size is small, and some block labels have no line breaks.

---

### Official Review · Reviewer_Nn5S · 2025-11-01

**Soundness:** 1
**Presentation:** 2
**Contribution:** 1
**Rating:** 2
**Confidence:** 5

**Summary:**

This paper proposes $\text{ScaleMoR}$, a multi-scale $\text{MoE}$ architecture for $\text{MTSF}$. It targets misalignment and inefficiency via three innovations: temporal-aligned tokenization, multi-dimensional complexity routing, and a recursive $\text{MoE}$ using shared linear layers at different depths. The paper claims $\text{SOTA}$ performance with massive reductions in parameters and $\text{FLOPS}$.My preliminary assessment is that this work is fundamentally flawed and unsound. The core methods are ill-defined, unsubstantiated by the provided equations, or mislabeled (e.g., calling hard-coded selection "adaptive"). The main $\text{MoE}$ concept also appears incremental. I recommend a Score 2 (Reject).

**Strengths:**

The fusion of $\text{KAN}$ with frequency decomposition is a strong concept for non-stationary $\text{TSF}$. The architecture is efficient, reducing $\text{MACs}$ and parameters versus Transformers while maintaining competitive accuracy. The method segments input into low, mid, and high-frequency components, capturing trends, periodicity, and abrupt changes. Ablations confirm $\text{KAN}$ superiority over $\text{MLPs}$ and highlight the $\text{ASFM}$'s importance.

**Weaknesses:**

W1. Fatal Flaw in Temporal-Aligned Tokenization (Soundness)This is a critical flaw. This tokenization is the paper's most important component, yet its description is non-functional. The paper claims alignment via Gaussian interpolation, defined as:$$W_{i,t}^{(s)}=exp(-\frac{(t-c_{i}^{(s)})^{2}}{2\sigma^{2}})/Z_{i}^{(s)}$$This weight $W_{i,t}^{(s)}$ is never used. Tokens are extracted via pooling $T_{i}^{(s)}$ and projected to $H^{(s)}$; the defined Gaussian weight is mathematically absent from this data flow. This is a fundamental gap between claim and method.

W2. Unsubstantiated Multi-Dimensional Complexity Routing (Soundness)The paper claims an estimator analyzes trend, seasonal, noise, and volatility. Equation 1 shows this is just an $\text{MLP}$:$$C=\sigma(W_{C}ReLU(W_{C1}h+b_{C1})+b_{C})$$No inductive bias forces the output vector $C$ to match these semantic labels. The model learns a 4D vector, but the labels are unsubstantiated.

W3. Misleading Definition of "Adaptive" (Contribution)The term "adaptive" is misused.The "Adaptive Prediction Head" is described as dynamic. The equation $y = ...$ reveals a hard-coded if-else statement based on prediction length $L \le 96$, $96 < L \le 336$, or $L > 336$. This is hard-coded selection, not adaptation.Similarly, the "horizon-aware" weights $w_{s}^{(L)}$ for $h_{global}=\sum_{s}w_{s}^{(L)}\cdot Pool(H_{processed}^{(s)})$ are fixed, non-learned values.Over-claiming "adaptivity" obscures the simpler, hard-coded design.

W4. Lack of Novelty in Core $\text{MoE}$ Concept (Contribution)The core $\text{MoE}$ claim is redefining experts as recursive depths. However, the paper's own related work section cites a framework that "applied shared layers recursively with dynamic depth". The paper fails to differentiate its $\text{MoE}$ concept from this prior art.

W5. Contradictory Training StrategyThe proposed three-phase training is contradictory. Analysis shows it hurts performance on the data-rich $\text{ETTm1}$ (14.6% $\text{MSE}$ increase) and only helps on the data-scarce $\text{ILI}$. A training method that fails on larger datasets is not robust.

**Questions:**

Q1. In which equation is the Gaussian weight $W_{i,t}^{(s)}$ applied to the tokens $T_{i}^{(s)}$ or $H^{(s)}$? It appears unused.

Q2. How does the $\text{MLP}$ (Eq. 1) semantically disentangle trend, seasonal, noise, and volatility from $h$ without an inductive bias?

Q3. In $R=Softmax(W_{R}[h;C]+c_{trend}B_{trend}^{(s)}+c_{seasonal}B_{scasonal}^{(s)})$, what are $c_{trend}$ and $c_{seasonal}$? If they are from $C$, why are $C_{noise}$ and $C_{volatility}$ ignored?

Q4. How does your recursive $\text{MoE}$ differ from the "Mixture-of-Recursions" framework cited in your related work?

---

### Note · Authors · 2025-12-25

I have read and agree with the venue's withdrawal policy on behalf of myself and my co-authors.